# Cooperation in a fluid swarm of fuel-free micro-swimmers

Matan Yah Ben Zion [1,2✉], Yaelin Caba[1], Alvin Modin[1] & Paul M. Chaikin[1]

While motile bacteria display rich dynamics in dense colonies, the phoretic nature of artificial micro-swimmers restricts their activity when crowded. Here we introduce a new class of synthetic micro-swimmers that are driven solely by light. By coupling a light absorbing particle to a fluid droplet we produce a colloidal chimera that transforms optical power into propulsive thermo-capillary action. The swimmers' internal drive allows them to operate for a long duration (days) and remain active when crowded, forming a high density fluid phase. We find that above a critical concentration, swimmers form a long lived crowded state that displays internal dynamics. When passive particles are introduced, the dense swimmer phase can re-arrange to spontaneously corral the passive particles. We derive a geometrical, depletion-like condition for corralling by identifying the role the passive particles play in controlling the effective concentration of the micro-swimmers.

[1] Center for Soft Matter Research, Department of Physics, New York University, 726 Broadway Avenue, New York, NY 10003, USA. [2] UMR Gulliver 7083 CNRS, ESPCI Paris, PSL Research University, 10 rue Vauquelin, 75005 Paris, France. ✉email: matanbz@gmail.com

ooperation is vital for the survival of a swarm[1]. Large scale cooperation allows murmuring starlings to outmaneuver preying falcons[2], shoaling sardines to outsmart sea lions[3], and homo sapiens to outlive their Pleistocene peers[4]. On the micron-scale, bacterial colonies show excellent resilience thanks to the individuals' ability to cooperate even when densely packed, mitigating their internal flow pattern to mix nutrients, fence the immune system, and resist antibiotics[5–14]. Production of an artificial swarm on the micron-scale faces a serious challenge— while an individual bacterium has an evolutionary-forged internal machinery to produce propulsion, until now, artificial micro-swimmers relied on the precise chemical composition of their environment to directly fuel their drive[14–23]. When crowded, artificial micro-swimmers compete locally for a finite fuel supply, quenching each other's activity at their greatest propensity for cooperation.

## Results

In order to explore the dynamics of a dense swarm of micro-swimmers, we synthesized a colloidal dimer by coupling a light-absorbing particle (M280) to a liquid droplet (n-dodecane), forming a 5 μm long, peanut-shaped swimmer (see Figs. 1a and 2, and Methods for synthesis steps[24]). When exposed to light, dimers commence a persistent swimming motion; when light is

turned off, the dimers revert to random Brownian motion (see Fig. 1b, c). As a light source, we used a wide beam (diameter $d = 310$ μm) infrared (wavelength $\lambda = 1064$ nm) laser (see Supplementary Information for details). Typical experiments were performed in deionized water, yet swimmers are also active in saline phosphate buffer (PBS), or when exposed to visible, non-coherent, light sources (see Methods for experimental setup and sample preparation). We find drastically different collective dynamics of the active particles when their area fraction, $\phi_A$, is increased, transitioning from quickly dispersing when diluting ($\phi_A < 0.14$, Fig. 1b, c and Supplementary Video 3); to forming transient, aligned aggregates at the intermediate concentration ($0.14 < \phi_A < 0.28$, Fig. 1d–g, and Supplementary Video 4); Finally, at high concentrations, swimmers crowd in a dense, active colony with internal flows ($0.48 \leq \phi_A$, Fig. 1h, i, and Supplementary Video 5).

In both the intermediate and high concentration regimes individual swimmers are often observed to swim within, and sometimes against, a group of other active particles (see Figs. 1d–f and 3a, and Supplementary Videos 1 and 4). This high degree of autonomy has been observed in living bacteria[9–14], but was not reported in previously studied systems of artificial micro-swimmers fuelled by either external ionic currents[15,16], consumption of micelles[20], hydrolysis of hydrogen peroxide[23], or

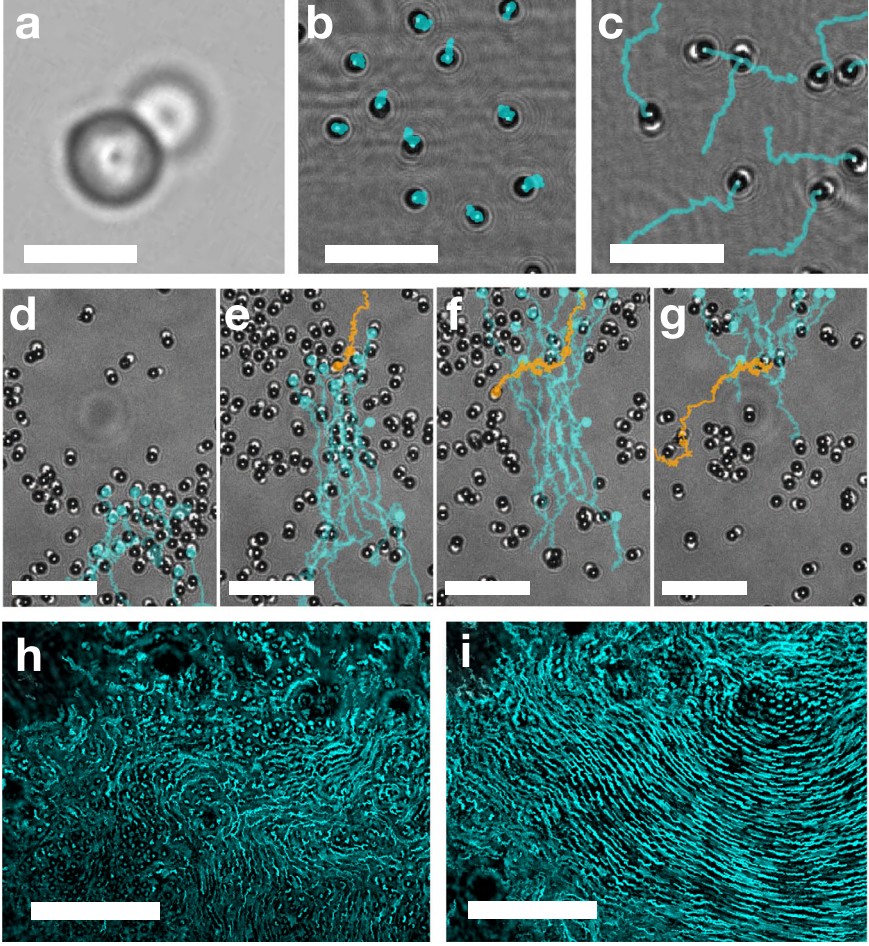

**Fig. 1 The light-driven fuel-free micro-swimmers display distinct behaviours at different densities. a** a single micro-swimmer is composed of a light-absorbing particle (dark sphere), and a fluid droplet (white sphere). 8 second long trajectories of individual swimmers show the transition from diffusive dynamics when light is off (**b**) to ballistic when light is turned on (**c**) (see Supplementary Video 3). **d** at an intermediate concentration, swimmers collide, align, and create small co-moving groups. **d–g** an individual swimmer (orange) maintains its activity even when passing through an oppositely moving group (cyan), penetrating, and emerging from the other side (see Supplementary Video 4). **h, i** long exposure (1 min) images of swimmers at high concentrations reveal internal flow structures (see Supplementary Video 5). Scale bars: **a**: 5 μm; **b**, **c**: 20 μm; **d–g**: 25 μm; **h, i**: 50 μm.

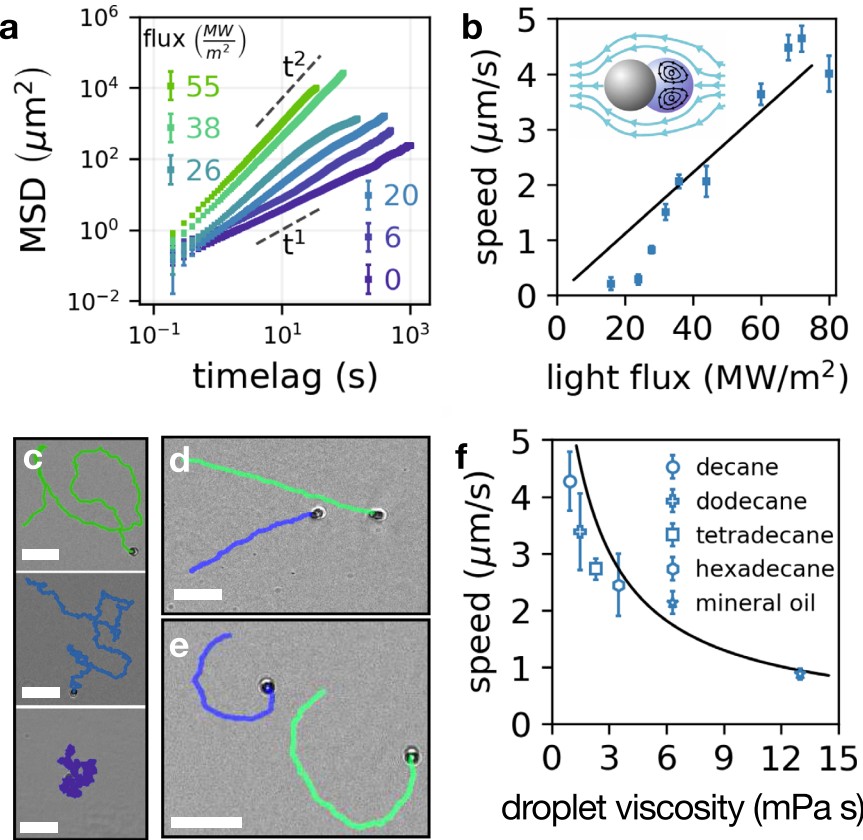

**Fig. 2 The internal drive of the fluid micro-swimmers relies on light-activated thermo-capillary action, independent from a chemical fuel in the surrounding bath. a** the mean square displacement (MSD) shows an individual swimmer's motility can be continuously tuned from diffusive ($\propto t^1$) to ballistic ($\propto t^2$). **b** the speed of a swimmer scales linearly with the power flux of the light source reaching more than one body length per second. Inset illustrates the fluid-flow where the surface-tension gradient drives the liquid–liquid interface to re-circulate (black) and propel the swimmer through the hydrodynamic coupling to the external flow (cyan). **c** with increasing power flux, trajectories of free swimmers show the transition from random (bottom) to more ballistic (top) motion. Using an external magnetic field the orientation of a swimmer can be controlled to move on a straight line (**d**) or in circles (**e**) allowing direct measurement of the nominal swimming speed. **f** increasing the internal viscosity of the fluid droplet directly reduces the speed of the swimmer. The solid line is the theoretically predicted swimming speed using no fitting parameters (full characterization of optical absorption and thermo-capillary measurements are found in Supplementary Information). Scale bars: 20 μm.

critical de-mixing of their surrounding buffer[17]. Instead, those systems exhibit interrupted internal dynamics, with the crowded phase being frozen and sometimes crystalline.

The propulsion mechanism of a single active particle is quantitatively captured using a thermo-capillary drive for a fluid droplet at a local temperature gradient[25,26]. Since surface tension, $\gamma$, is a powerful agent on the micro-scale and is generally temperature dependent[27,28], $\gamma = \gamma(T)$, a small local temperature-gradient suffices to generate considerable propulsion. Using light, we heat up the light-absorbing particle to establish a local temperature gradient, $\overrightarrow{\nabla} T$, at the vicinity of the fluid droplet. The temperature gradient generates a surface-tension gradient proportional to the thermo-capillary coefficient, $\beta \equiv \frac{\partial \gamma}{\partial T}$. The velocity, $\overrightarrow{v}$, of a fluid droplet in a temperature gradient is given by[25]:

$$\overrightarrow{v} = -\left[\frac{1}{2\eta_f + 3\eta_w}\right] \frac{D}{2 + \kappa_w/\kappa_f} \beta \overrightarrow{\nabla} T, \quad (1)$$

with D being the droplet diameter, $\eta_i$ is linear viscosity, and $\kappa_i$ the thermal conductivity (index $i$ indicates $w$-ater or $f$-luid droplet). Note that for high droplet viscosity, $\eta_f \gg \eta_w$, the swimming speed is dictated by the internal flow inside the fluid droplet, $v \propto \beta \nabla T/\eta_f$. We independently measured both the heating of the light-absorbing particle and the thermo-capillary coefficient of the fluid droplet, finding numerical values consistent with existing

literature[29–31] (see Supplementary Information). By synthesizing a family of swimmers made from a homologous series of alkanes, we observe a decrease in swimming speed, $v$, with increasing internal viscosity, $\eta_f$, as required by a thermo-capillary mechanism (Fig. 2f)[32]. When accounting for the reduced mobility of particles near a solid surface[33], we find quantitative agreement between measured swimming speed, and the light-driven thermo-capillary model for swimmers of different compositions (see Fig. 2f) with no fitting parameters.

An individual swimmer can be oriented using an external magnetic field (through the small remanent moment of the superparamagnetic light-absorbing particle) as shown in Fig. 2d, e. Fixing the orientation allows direct measurement of the swimming speed. Decoupling propulsion from random motion we find a persistent time of $\tau \approx 15$ s for the free swimmer (Fig. 2a and Supplementary Information). With increasing light intensity, individual swimmers transition from Brownian to ballistic (Fig. 2a) and can move up to 3 body lengths per second (15 μm/s). Here we focus on the slower swimming speeds, $v \leq 5$ μm/s, corresponding to a Péclet Number, $P_e$, of up to 200. In the presence of other swimmers, the individual swimmer remains active, but with dramatically different dynamics.

The individual swimmer motility is characterized by a significant slowdown once entering a region of high local concentration (see Fig. 3a and Supplementary Video 1). It is known

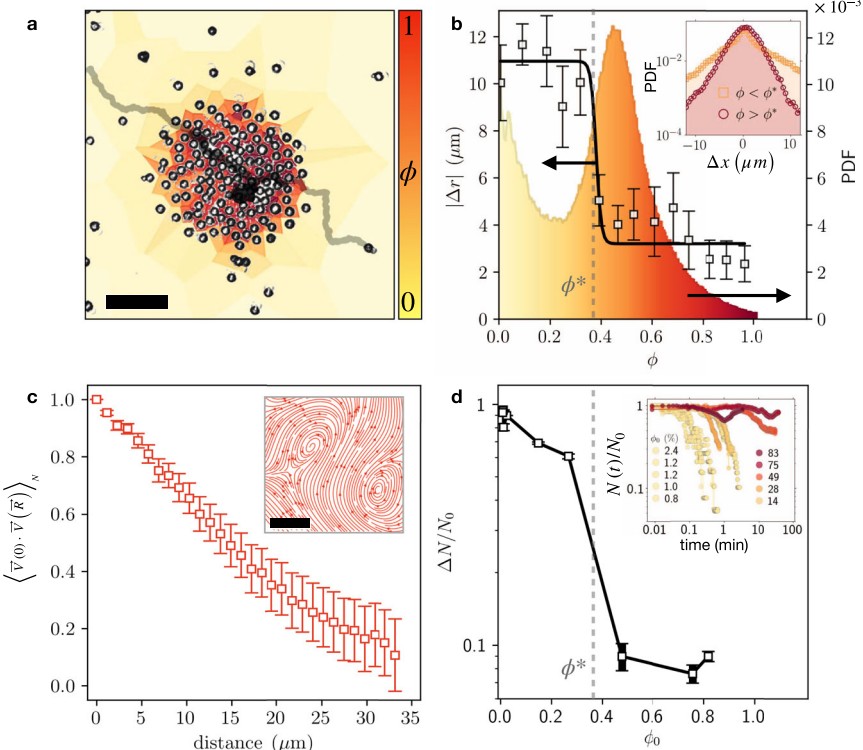

**Fig. 3 At sufficiently high average area fraction, $\phi_O$, the swimmer population segregates into a long-lived state of dilute and dense fluids. a** The instantaneous local density of each swimmer is measured from the inverse size of the corresponding Voronoi cell. Both the dilute and the dense phases are fluid as can be seen from the trace of an individual swimmer crossing the different phases, moving slower (darker trace) in the denser region (see Supplementary Video 1). **b** the probability for finding an individual swimmer at a specific local density shows two characteristic phases. Particle displacement over a fixed interval (5 seconds) shows a sharp decrease above the local concentration of $\phi^* = 0.38$ (black squares). Inset shows the displacement probability distribution function for an individual swimmer below (yellow squares) and above (red circles) $\phi^*$, showing that the ballistic motion at low density has a PDF with heavier tails than the PDF of the collision dominated displacement at high densities. **c** velocity correlation function within the dense phase indicates flows correlate on a length scale much larger than the swimmer size. Inset shows streamlines measured using particle image velocimetry, displaying an internal flow structure consistent with the swimmers slightly repulsive nature (see Supplementary Information). **d** at lower initial concentration, $\phi_O \le 0.28$, most of the swimmers will disperse, $\Delta N \gg N_O$, within a few seconds up to minutes. At higher initial concentrations, $\phi_O \ge 0.49$, the swarm forms a persistent crowded state, which remains throughout the measurement time (hours). Inset shows temporal evolution. Scale bars are 20 μm.

that the speed-concentration dependence of the individual swimmer, $v(\phi)$, can be used to predict the critical concentration, $\phi^*$, at which the whole swarm tends to form a persistent dense region[34]. The trajectory of a single swimmer as it samples a heterogeneous density field is shown in Fig. 3a and Supplementary Video 1. The long-lived crowded state is characterized by the co-existence of dense and dilute regions, with corresponding peaks in the density probability distribution (Fig. 3b). By measuring the local, instantaneous area fraction, $\phi$ (using Voronoi tesselation of surrounding swimmers, see Supplementary Information), we find that a swimmer maintains its nominal speed at low concentrations, and experiences a sharp decrease in motility inside the dense region (Fig. 3b). The swimmer regains its nominal speed once emerging from the other side of the dense region. It is theoretically predicted that when the decrease in speed with concentration is sufficiently sharp, $\frac{\partial v}{\partial \phi}|_{\phi=\phi^*} \le -\frac{v}{\phi}$, concentration gradients appear at a steady state in a process known as Motility Induced Phase Separation[34]. The critical concentration we anticipate from $v(\phi)$ if Fig. 3b is $\phi^* \approx 0.38$, consistent with numerical predictions found in the literature[34–39]. Indeed, the swarm's dynamics change dramatically above this critical concentration. Below $\phi^*$ the initial population, $N_0$, quickly disperses, with the lion's share of the swimmers, $\Delta N$, leaving the observed field of view $\Delta N \approx N_0$. By contrast, at concentrations

above $\phi^*$ the swarm maintains a long-lived crowded huddle, with only a small fraction of the swimmers leaving, $\Delta N \ll N_0$ (see Fig. 3d). As in dense bacterial colonies[7,10,13], the crowded swimmer phase is liquid, characterized by flow correlations much larger than the individual. Even crowded regions of over 1000 swimmers display a turbulent flow (Fig. 3c), allowing internal re-arrangement, a key ingredient for their cooperativity.

As the swimmers remain active even at high concentrations, they cooperate to corral passive particles (Supplementary Video 6). Figure 4b–d shows the time evolution of a heterogeneous mixture of active and passive particles, where the passive particles are compressed by active particles into a dense phase with hexagonal order, as seen in the pair correlation function, $g(r)$, and the structure factor, $S(\vec{q})$ in Fig. 4b–e (see Supplementary Information). Microscopically, the corralling builds up through a series of entrainment events, similar to those observed with microalga[40] where active particles chaperon passive particles to the dense region. The passive particles are then deposited while the active particles depart (Fig. 4f and Supplementary Video 2). Dynamically, the corralling process proceeds through two stages: **1.** at early times, when the dense, passive-active, suspension is still well mixed (Fig. 4b), the ability of the swimmers to re-arrange when crowded allows them to escape the dense region (Fig. 4c, d. also see Supporting Video S6); **2.** at later times, individual

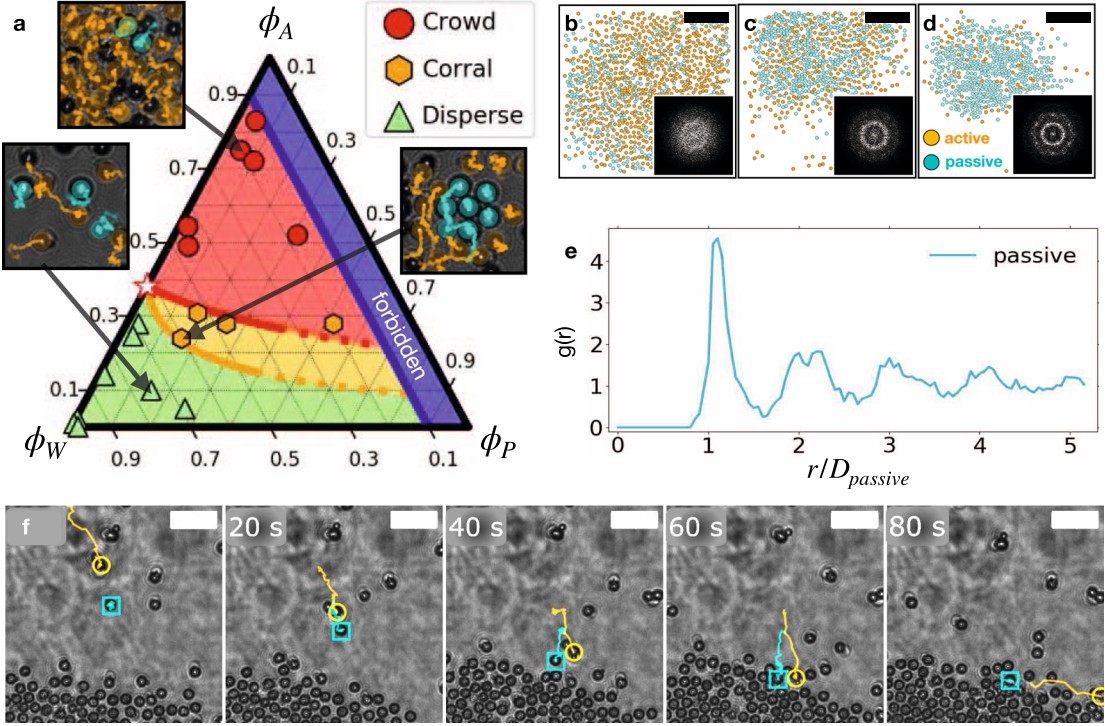

**Fig. 4 A swarm of active swimmers cooperatively corrals passive particles. a** a ternary phase diagram of active ($\phi_A$) passive ($\phi_P$) and the surrounding water bath ($\phi_W$) depicts the resulting dynamics of a mixture of passive particles and active swimmers, given their initial area fraction. Green triangles show that at low overall-particle area fraction (large $\phi_W$), the active swimmers quickly disperse. Red circles show a region where the active particles crowd once activated and form a persistent dense phase. Between the red and the green regions, $1 - 4\phi_P < \phi_A/\phi^* < 1 - \frac{\sqrt{12}}{\pi}\phi_P$, a yellow region emerges (see Eq. (2)), where orange hexagons signify experiments where active particles spontaneously corral the passive particles ($\phi^*$ is denoted by a star). Red and orange curves are theoretical predictions for the onset of the corralling state extrapolated for the small $\phi_P$ limit. Thumbnails show snapshots from corresponding points on the diagram (orange and cyan label active and passive particles, respectively). Blue region indicates an overall-particle area fraction greater than 0.91 (hexagonal packing). **b–d** snapshots taken at time 0 min (**b**) 5 min (**c**), and 30 min (**d**) show the global dynamics of the active particles (orange) as they corall the passive particles (cyan) (see Supplementary Video 6). Insets are the structure factor, $S(\vec{q})$, of the passive particles, showing the evolution towards hexagonal order in their packed arrangement. **e** the pair correlation function has peaks at 1, 2, 3, and 4 diameter, with $1 - \sqrt{3}$ splitting around 2 indicative of ordered hexagonal packing. **f** snapshots at 20 seconds intervals show the corralling process which proceeds through entrainment events, where an active particle (orange) deposits a passive particle (cyan) at the dense region. After deposition, the active particle leaves (see Supplementary Video 2). Scale bars: **b–d** 30 μm; **f**: 10 μm.

entrainment events are observed (see Fig. 4f and Supporting Video S2).

The resulting dynamics observed for different initial area fractions are summarized in a ternary phase diagram (Fig. 4a). We note that the area fraction of the active particles $\phi_A$, passive particles, $\phi_P$, and the surrounding water phase, $\phi_W$ add up to unity $\phi_A + \phi_P + \phi_W = 1$. When the initial concentration is less than dense packing (blue region $\phi_A + \phi_P \leq 0.907$), we identify three characteristic behaviours: active particles quickly disperse at low particle area fraction (green region); a long-lived dense active state is formed at higher concentrations (red region); at intermediate particle concentrations, the active particles corral the passive particles (yellow region).

A simple geometrical construction captures the corralling phase boundaries in the phase diagram. Introducing passive particles reduces the total area available for the active particles, increasing their effective area fraction, $\phi_A^{\text{eff}}$. The increased effective area fraction may surpass the critical concentration, $\phi^*$, and trigger the crowding dynamics. The extent of this effect, however, depends on the spatial arrangement of the passive particles. When randomly dispersed each passive particle occupies much more room as it carries a corona of excluded area. When the passive particles are packed in a lattice (as in Fig. 4d, e), their excluded areas overlap and they effectively occupy a smaller area. Therefore we expect the effective area fraction of the active

particles to reduce when the passive particles are hexagonally packed (HP), $\phi_A^{P,\text{rand}} > \phi_A^{P,\text{HP}}$. For circular passive particles in 2D we can estimate the effective area fraction of the active particles for the two cases as,

$$\phi_A^{\text{eff}} \approx \begin{cases} \phi_A/\left(1 - 4\phi_P\right) & \text{passive are random} \\ \phi_A/\left(1 - \frac{\sqrt{12}}{\pi}\phi_P\right) & \text{passive are HP} \end{cases} \quad (2)$$

(see Supplementary Information for full derivation). The red and orange curves in Fig. 4a, respectively, represent the point at which the effective area fraction of the active particles is equal to their critical area fraction for crowding, $\phi_A^{P,\text{rand}} = \phi^*$ and $\phi_A^{P,\text{HP}} = \phi^*$. The corralling region is found between those two curves.

## Discussion

The following picture arises: when active and passive particles are homogeneously mixed, the passive particles can trigger the crowding of the active particles at concentrations lower than the critical concentration of the pure active system, $\phi^*$. An increased concentration of active particles reduces their motility, and past a critical concentration ($\phi^*$) triggers MIPS. Similarly, an increased concentration of passive particles also reduces the motility of the active particles, which in turn also leads to clustering instability. The effective increased concentration, however,

depends on the spatial arrangement of the passive particles. Once the active particles complete the corralling task, they disperse, as the area gained by the overlapping excluded areas makes them sub-critical. A similar equilibrium process is the depletion interaction[27,41], where the diffusion of many small particles (depletants) can drive fewer, larger particles to crystallize. There is one important distinction: even in the absence of passive particles, active particles have an intrinsic critical clustering concentration because of their activity. Previous numerical simulations suggested that at low concentrations, active particles may induce an effective interaction between a pair of passive particles through what is known as Active Casimir Effect[42–44]. Here our analysis shows qualitatively similar results to previous theoretical work predicting phase separation in mixed active/passive populations[36,37,45–47], but may prove more general as it relies on a generic geometrical argument.

In this work, we have introduced a synthetic route and a driving strategy for a new family of micro-swimmers that are propelled solely by light—a colloidal chimera of a light-absorbing particle bound to a fluid droplet. An individual swimmer's propulsion mechanism is quantitatively captured by a thermo-capillary drive. We demonstrate that the synthetic route makes swimmers of different materials accessible, with tuneable fore-aft geometrical asymmetry, making previous numerical work experimentally realizable[45]. We find that the swimmers remain active at high area fractions, $\phi_A > 0.5$, and can be used as an experimentally tuneable model system to study a host of phenomena in dense active matter including 2D turbulence in a compressible fluid, previously seen in bacterial colonies[9–13] and in groups of ants[48,49]. We found that at a low area fraction ($<0.15$), swimmers tend to quickly disperse; at intermediate area fractions (0.14–0.28) swimmers form transient co-moving clusters, reminiscent of the extensively studied flocking starlings, and schooling golden shiners[50,51]. Above a critical concentration ($\phi^* \approx 0.38$), the micro-swimmers form a long-lived dense phase. We identify a depletion-like interaction where the excluded volume of passive particles depends on their geometrical arrangement (random or packed) and can change $\phi^*$ of the active particles resulting in passive particle crystallization. As the mechanism is geometric, it may be applicable generally from living organisms to robotic swarms. As our optically powered Marangoni swimmers are largely agnostic to their precise chemical environment, we were able to graft DNA on their surface. Micro-swimmers augmented by DNA nanotechnology[52–54] can prove useful in the self-assembly of active tissue, mimicking biofilms, in the making of future, *functional* active matter.

## Methods

### Swimmer synthesis and sample preparation

*Synthesis overview.* Hybrid dimers were synthesized by binding a 3 μm light-absorbing particle (M280 BANGS LAB) to a 3 μm oil droplet (see list in Supplementary Information) using the previously introduced Mix-And-Match strategy[24]. The following describes the synthesis of swimmers made of M280 ($\rho_{M280} = 1.39\,g/cc$) coupled to a fluid droplet made of n-Dodecane ($\rho_{nC12} = 0.75\,g/cc$), with dimer having a composite density of $\rho_{dimer} = 1.07\,g/cc$ (see Supplementary Information).

*Emulsion synthesis.* Fluid droplets were made through membrane emulsification with a typical droplet size of $3.0 \pm 0.6$ μm, using oil with 0.1% v/v SPAN80 (SIGMA Aldrich). 10 mL oil loaded into an emulsification cup is pressurized at ~200 $kPa$ through a 0.5 μm porous membrane (SPG technology), and dipped into a beaker with 100 mL deionized water continuously stirred with a magnetic stirrer forming an oil in water emulsion. After emulsification, the suspension is loaded into a separation funnel for 1–2 h, followed by extraction (thus removing the larger, more buoyant, droplets). After 2 separation steps, the particle size was 3 μm. The suspension is then stored in a glass vial and is stable as a dense cream at room temperature for months. The process can be repeated for improved monodispersity. Note that a quicker way to make an emulsion (with lower throughput

and purity) is accessible using sonication where 0.4 mL oil (with 0.1% w/w SPAN 80) and 0.6 mL water are vortex mixed in a mini-centrifuge tube and then sonicated in a sonication bath for 5 min.

*Coupling light-absorbing particles to liquid droplets.* Typically 50 μL of 2.3%w/w M280 3 μm dynabeads are mixed into 3 mL of DI water inside a 50 mL falcon tube, and sonicated for 10 min. Then 10 μL of the dense emulsion cream (60%v/v) is diluted by pipette mixing into 990 μL DI water in a mini-centrifuge tube, followed by the introduction of the whole content of the centrifuge into the Falcon tube and pipette mixed (see Supplementary information) giving a roughly stoichiometric mixture of M280 and droplets. The suspension is then destabilized by increasing its ionic strength where 250 μL NaCl 0.1 M are added to the falcon tube and pipette mixed for 30 seconds, quickly followed by a reaction quench by adding 40 mL 0.1% F127 in DI into the reaction tube to re-stabilize the suspension. The Falcon tube is then tumbled on a rotating arm for 40 min. This is followed by a wash cycle where the Falcon tube is centrifuged at 1kRPM in (IEC centraCL2 centrifuge) for 20 min, and decanted, where typically 5–6 mL suspension is left.

*Dimer purification.* Dimers are purified from the suspension using isopycnic density gradient step centrifugation[24,55]. A transparent plastic centrifuge tube (Seton Open top Polyclear 14 × 89 mm) is loaded with the suspension (6 mL), followed by injection of 1 mL 1.026g/cc glucose solution to the bottom of the tube (using a long pipette, and another layer below of 1 mL, 1.077 g/cc glucose solution (see Supplementary Information). The layered tube is loaded into a swinging bucket centrifuge (Marathon 6 K), and spun at 4 kRPM for 25 min. A band is formed between the two sucrose phases, which is readily extracted using a needle syringe (BD Precision Glide Needle 27GX1/2) making a 0.5 mL transparent suspension. The suspension is transferred to a mini-centrifuge tube and centrifuge washed with DI at 1.5 kRPM for 20 min (Eppendorf Minispin 2) to remove excess sucrose. A visible, brown pellet is found after centrifugation. Two hundred microlitres DI is added to the suspension which is then stored in a 4 °C refrigerator and found to be stable for months.

*Sample preparation.* Samples were prepared for imaging by gentle tumbling of the purified dimers to resuspend the particles. Then ~10 μL are pipetted into a pre-treated glass capillary (channel height 0.10 mm, width 2.00 mm Vitrotubes W5010-050), which were plasma cleaned (SPI supplies Plasma Prep II), and passivated through vapour deposition of hexamethyldisilazane (SIGMA). Loaded capillaries were placed on a clean microscope glass slide, and sealed on their ends with UV curable resine (LOON OUTDOORS UV CLEAR FLY FINISH).

**Experimental setup.** Imaging was done using bright-field on a home-built microscope coupled to a laser source. A commercial light-emitting diode ($\lambda = 505$ nm THORLABS) with a diffuser (ground glass N-BK7 600 grit, THORLABS), a condenser and an iris, to achieve Köhler illumination. The scattered light was picked up by the microscope objective (HCX PL APO ×40 NA = 0.85, LEICA), and a tube lens (B&H), and detected by a digital camera (DCC1545M, IMAGING SOURCE) and acquired using commercial video recording software (IC CAPTURE, IMAGING SOURCE). A heating beam was introduced on a separate optical path (see Supplementary Information). A 1064 nm laser beam (YLR-10-1064-LP, IPG Photonics) was passed through a zero-order half-plate (WPH05M-1064 Thorlabs) and contracted using a custom Galilean telescope to achieve a ~300 μm beam (see Supplementary Information). The laser beam was introduced into the sample using a polarizing beam splitter (PBS CM1-PBS253 THORLABS) and its intensity at the sample was controlled by a combination of the electronic laser head controller and by adjusting the half-plate, and was measured using an optical power meter (PM100D power meter, with S175C sensor, THORLABS). In order to eliminate laser intensity before the camera, stained glasses (FGS900S, THORLABS) were stacked after the objective.

## Data availability

Data are available at an online repository[56] in the following URL: https://doi.org/10.6084/m9.figshare.16559733.v1.

## Code availability

The custom codes used in this study are available from the corresponding author upon request.

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

## Acknowledgements

We thank N. Oppenheimer, O. Dauchot, A. M. Leshansky, T. Goldfriend, and C. P. Kelleher for insights. This research was primarily supported by the Center for Bio-Inspired Energy Sciences, an Energy Frontier Research Center funded by the DOE, Office of Sciences, Basic Energy Sciences, under award no. DE-SC0000989), and the Diversity Undergraduate Research Initiative (DURI) New York University.

## Author contributions

M.Y.B.Z. and P.M.C. planned the experiments. M.Y.B.Z. and Y.C. developed the synthesis. M.Y.B.Z. and A.M. built the experimental setup. M.Y.B.Z. conducted the experiments, analyzed the results, and developed the theoretical model. M.Y.B.Z. and P.M.C. wrote the paper.

## Competing interests

The authors declare no competing interests.
