## [Peer Review File · Nature Communications]

REVIEWER COMMENTS

Reviewer #1 (Remarks to the Author):

In this manuscript special artificial peanut-shaped microswimmers are designed which possess special features. Two main findings are put forward. First individual particles exhibit a large persistence when penetrating through a swarm of other particles, hence they maintain largely their self-propulsion velocity. Second, in a mixture of active and passive particles, the passive particles form a cluster which is encircled by active particles. The latter phenomenon is explained in terms of a modified depletion picture.

The paper is written in a clear way.

Given the fact that in the field of active matter there are only few quantitative experiments (as compared to a relatively high number of simulation papers), the manuscript deserves appreciation such that Nature Communications could be the right place to publish it.

However, I have two major concerns 1) and 2) regarding the basic message which the authors should explain.

1) On page 1, the argument is brought that artificial micro-swimmers compete locally for a finite fuel supply which in a certain sense distinguish them from motile bacteria. However, in principle, the same competition applies here for the peanut-shape swimmers. Just fuel supply is replaced by temperature gradient. In more detail, it is argued on page 7, that the temperature field around the heated particle roughly decays as $1/R$ and this is very long-ranged. When in a fluid swarm there are many neighboring temperature sources which reduce the local temperature gradient around the droplet such that the motion of the colloidal micro-swimmer should be reduced. Or in other terms: also a fuel source/sink (for chemo-tactic response) should decay as $1/R$ such that there is no difference in the physics. I rather speculate/suggest that the physical reason for the penetration of the particles is a strong alignment induced by the particle shape such avoiding central collision which would impeded particle motion for spherical particles. Therefore this alignment should be analyzed in a more quantitative manner.

2) The idea of effective attraction between passive particles in an active bath is not new. There have been several investigation on activity-induced depletion interactions: D. Ray et al, Phys. Rev. E 90, 013019 (2014); J. Harder et al, J. Chem. Phys. 141, 194901 (2014); F. Smallenburg et al, Phys. Rev. E 92, 032304 (2015). These papers should be a least mentioned. They also show that depletion interactions can be repulsive which would destroy the picture proposed by the authors. Can the authors measure the effective force between a pair of peanut particles to see the basic effect on the particle pair scale? While I agree that the depletion picture is a good one, it should be supported with more data.

3) In the methods part it is stated that dimers were synthesized by binding a light absorbing particle to a fluid oil particle. The mechanisms of binding should be explained in more detail.

4) The insets of Figure 3B and 3D are invisible. I wonder why the displacements in 3B for $\phi < \phi^*$

are smaller than that for $\phi > \phi^*$ but maybe this is due to the poor resolution of the figure.

5) In the movies trimers and spinners are seen, is there any comment on those?

6) typo on page 3: Suuplementary

Reviewer #2 (Remarks to the Author):

The authors introduce a novel type of synthetic polar active particle by coupling a light absorbing colloid to a fluid droplet, resulting in a surface-tension driven propulsion mechanism. In contrast to chemically powered active colloids, this new experimental microswimmer system enables the realization and study of long-lived coherent non-equilibrium dynamics at high density.

The authors explain and characterize the dynamics of individual swimmer in terms of a thermocapillary model, and they observe interesting collective dynamics at intermediate and high filling friction. The authors further analyze mixtures of active and passive colloids, and find that their experimental observations agree with theoretical predictions based on simple geometric considerations (Eq.2, Fig. 4)

The paper is well written, the experimental characterization is comprehensive, and the theoretical arguments appear convincing.

More generally, I believe that this new synthetic active particle system provides an interesting alternative to chemically powered colloidal systems. I am therefore happy to recommend this work for publication, but would ask the authors to consider and address the minor questions below.

Specific questions/comments:

Fig 1B,C: What is the duration of the trajectories shown in these two panels, respectively?

2. Fig. 3: The segregation into a long lived state of coexisting dilute and dense fluid phase seems reminiscent of the phenomenology predicted in Fig. 2(d) of Phys Rev E 89: 010302(R), 2014.

3. I am not entirely convinced by the "cooperation angle" of the presentation. The interesting collective dynamics reported in the paper arises mostly from basic physical interactions that do not seem to meet the criteria of "cooperation" in more complex biological systems.

Minor comments:

In brackets below Eq (1): remove "droplet"?

Ref [2] has wrong title and page number

Ref [5] has incorrect page/article number

Ref [13] has incorrect page/article number

Ref [18] has incorrect page/article number

Refs [33-36,39,41] have incorrect page/article number

Reviewer #3 (Remarks to the Author):

The authors present a new type of microswimmers that is based on a fuel-less non-phoretic propulsion mechanism, where the particles are supplied with energy by light.

The main advantage of the new propulsion mechanism is that the particles remain motile also in highly crowded assemblies.

Furthermore, the authors realized the particles in experiments and found that they show an interesting cooperative motion with parallels to swarming of animals.

The findings are certainly interesting for the large active matter community. A type of active particles that behaves more like bacteria or animals than the various phoretically propelled active particles is indeed useful. Artificial active particles are quite often compared to motile microorganisms and animals and these new particles could probably be applied as models for bacteria and animals.

The paper is well written and the presentation is clear.

However, I cannot recommend publication of the paper in its present form.

The authors should address the issues listed in the following.

Some problems can easily be solved by the authors. Other issues require to provide more information in the manuscript on which basis it will become clear whether the findings are sufficiently relevant for publication in the chosen journal.

1) The propulsion mechanism of the proposed particles should be explained in more detail. It would be very helpful if the authors provide a sketch illustrating clearly how it works.

2) It seems that a temperature gradient caused by illumination and a resulting flow field are crucial for the propulsion of the particles. How do they differ from the existing thermophoretically propelled particles studied, for example, by Cichos (Leipzig)?

3) Second paragraph, line 4: Replace "homosapiens" by "homo sapiens".

4) The authors mention collisions of the particles. The particles, however, consist of a combination of a solid particle and a droplet. Why do they stay intact after a collision? Why do the droplets not coalesce?

5) The authors should comment in more detail on how the motion of the particles compares to the cooperation of microorganisms and animals. To which extent are they similar?

6) Is equation (1) really applicable? It describes the velocity of a fluid droplet in a temperature gradient, but is the temperature gradient caused by heating of the small solid component of a particle really significantly larger than the particle? If yes, why is thermophoresis not relevant, which would make the particles just to another type of phoretic particles?

7) The figures should be improved, the quality is only average. The largest problem is the size of the labels. Sometimes they are too small and sometimes they are too large. For example, in figure 2A the numbers at the frame are much too small and in figure 3 the labels "A", ..., "D" are much too large.

8) Figure 3B, inset: Why are there two orange curves that look rather different and not just one curve?

9) Figure 3B: Here are a histogram and a curve plotted, but there is only one vertical axis. This is strange. In particular, the histogram seems to be a probability distribution and then it makes no sense that the vertical axis has the unit μm .

10) Figure 4A: The colors red, yellow and green are explained in the figure, but the legend should explain also what the blue region means. (I know that it is explained in the caption but this is not sufficient.)

11) Figure 4, caption, line 5, middle: Where does this equation come from?

12) Page 5, left column, paragraph 2, lines 2-5: I do not really understand the argument why the passive particles facilitate crowding of the active particles. It is true that they reduce the space that is available for the active particles, but on the other hand the active particles collide with the passive ones and slow down by these collisions so that the typical crowding mechanism of active particles (motility-induced phase separation) should no longer work.

13) The authors write that the particles form a dense fluid in the interior of the clusters and that they do not form a solid as it has been observed for phoretically propelled active particles. Does this also apply to very large clusters?

In Brownian dynamics simulations of active Brownian particles that can be found in the literature, it has been observed that the particles form a solid and even crystals in the interior of a cluster. This crystallization occurs although the propulsion mechanism of these particles is by definition not affected by the presence of other nearby particles.

Therefore I would expect that also the particles studied in the present paper show solidification in the clusters if they are sufficiently large.

14) The journal name of reference [15] is wrong.

15) Reference [22] is incomplete.

16) Acknowledgments, line 6, middle: ", and), and" is wrong.

17) Table 1 does not look good. At least a horizontal line between rows 1 and 2 is needed.

Reviewer #4 (Remarks to the Author):

This paper reports the development of a new class of colloidal microswimmers that promise access to dense regimes of active fluids that have heretofore not been able to be accessed in synthetic systems. The propulsion in the high density regime is accomplished by using light to propel the particles rather than a fuel that would be diffusion limited in crowded environments. Further this allows nice control of local velocity as a function of temperature gradient. Several well known results from the self-propelled particle literature are nicely validated here including MIPS. The active-passive phase separation system is underexplored but serves as a further validation of the potential for this system to explore the phenomenology of self-propelled particles in a well controlled system. I like it and have no criticism for the authors. All the active matter people will be looking forward with bated breath to further work on this system.

Response to Referees for Manuscript NCOMMS-21-18131-T

September 1, 2021

Response summary

Below is the response to reviewers for our manuscript entitled “*Cooperation in a fluid swarm of fuel-free micro-swimmers*” (manuscript number NCOMMS-21-18131-T). Our response is structured as follows: We first address points shared by multiple reviewers. This includes poor image resolution of some of the figures in the original version of the manuscript. This is followed by a point-by-point response to the comments, separated for each reviewer. Reviewers’ questions are highlighted in **bold**. Changes in the text are mentioned in the response, and highlighted in **green** in the new version of the manuscript. The biggest change in the updated version was made following a request by Reviewer’s #1 for a more elaborate description of the synthetic protocol, where we added four sections and four figures in the Methods.

Response to reviewers' comments

General Comments

Both reviewer 1 and 3 pointed out that labels in the insets of a few of the figures suffer from poor resolution which lead to confusion. To address their concerns we made the following changes:

1. Increased the font of the legend of the inset in Fig.3b.
2. Moved the right vertical axis label of Fig.3b to the right of the axis (and outside of the plot, as in the rest of the manuscript).
3. Added ticks to the right vertical axis in Fig.3b.
4. Added a left pointing arrow on the displacement curve in figure 3b, illustrating it belongs to the left axis.
5. Added a right pointing arrow on the histogram in figure 3b indicating it belongs to the right axis.
6. Changed the inset scaling to semi-log (y), which better illustrates the gaussian step size at high concentration and the heavier tails at low concentrations.
7. Increased marker size in the legend of Fig.3b.
8. Removed the gaussian fits in the inset of Fig.3b.
9. We updated the caption in Fig. 3b as follows :
“Inset shows the displacement probability distribution function for an individual swimmer below (yellow squares) and above (red circles) ϕ^* , showing that the ballistic motion at low density has a PDF with heavier tails than the PDF of the collision dominated displacement at high densities.”
10. Shaded the area under the PDF curves in the inset of Fig.3b.
11. Splited the legend in Fig.2a into two columns and increased the font size.

Reviewer #1

In this manuscript special artificial peanut-shaped microswimmers are designed which possess special features. Two main findings are put forward. First individual particles exhibit a large persistence when penetrating through a swarm of other particles, hence they maintain largely their self-propulsion velocity. Second, in a mixture of active and passive particles, the passive particles form a cluster which is encircled by active particles. The latter phenomenon is explained in terms of a modified depletion picture.

The paper is written in a clear way.

Given the fact that in the field of active matter there are only few quantitative experiments (as compared to a relatively high number of simulation papers), the manuscript deserves appreciation such that Nature Communications could be the right place to publish it.

We thank the reviewer for finding the manuscript well written, and endorsing its importance.

Response to comments

1. On page 1, the argument is brought that artificial micro-swimmers compete locally for a finite fuel supply which in a certain sense distinguish them from motile bacteria. However, in principle, the same competition applies here for the peanut-shape swimmers. Just fuel supply is replaced by temperature gradient. In more detail, it is argued on page 7, that the temperature field around the heated particle roughly decays as $1/R$ and this is very long-ranged. When in a fluid swarm there are many neighboring temperature sources which reduce the local temperature gradient around the droplet such that the motion of the colloidal micro-swimmer should be reduced. Or in other terms: also a fuel source/sink (for chemo-tactic response) should decay as $1/R$ such that there is no difference in the physics. I rather speculate/suggest that the physical reason for the penetration of the particles is a strong alignment induced by the particle shape such avoiding central collision which would impeded particle motion for spherical particles. Therefore this alignment should be analyzed in a more quantitative manner.

The reviewer correctly points out that unlike previous, chemically driven, micro-swimmers here “fuel is replaced by temperature gradient”. And indeed the particles in principle interact through the thermal gradient. It is however important to point out a couple of differences:

- (a) Chemically driven swimmers depend on the *absolute chemical concentration* of both the fuel and the reaction products, leading to a Michaelis-Menten like suppression of motility (see for example Ref 18). A phoretic swimmer changes the local concentration of reactants and products, leading to a slowdown or even a complete stopping of a nearby swimmer. By contrast, in our case, the energy source of the swimmer (light) is coming from “above”, and swimmers do not obscure the light from one another.
- (b) The reviewer correctly suggests that the swimmers do interact through the temperature field each produces (that scales like $1/R^2$, see Eq. 3 in section 1.1 in supporting information). However the material properties used in our design ensure that swimmer-swimmer thermal interactions are kept to minimum: The thermal conductivity of the surrounding water ($\kappa_{H_2O} = 0.6W/m \cdot K$) and adjacent cover slip ($\kappa_{glass} \approx 1.4W/m \cdot K$) are 4-10 greater than that of the oil ($\kappa_{oil} \approx 0.14W/m \cdot K$, see Table 1 in Supporting Information). This sets the greatest thermal gradient across the tightly coupled oil droplet.

The reviewer also suggests that the elongated shape of our micro-swimmers is important for their activity when crowded. We indeed observe the effect of alignment to play a role at a narrow range of intermediate concentrations. This has been described in the manuscript to lead to transient co-moving clusters (see Fig.1 d-g and Supplementary Video S4). Previous work on living micro-swimmers has also shown that the aspect ratio is important in the fluidity of their dense clusters (see Ref 6). It is however important to note that previous work on synthetic, fuel-driven, micro-swimmers did not show activity in dense clusters even for elongated rods. Example for such are catalytic bimetallic nanorods which were found to “form permanent static clusters” (see I. Aranson in Ref 10). To improve the manuscript, we highlighted the equation for the gradient (Eq. 3 in the updated version of the manuscript, previously inline) and added the following comments in the text:

- (a) The temperature gradient is given by the spatial derivative of the temperature profile around the hotter light absorbing particle.
- (b) On the length scale of the swimmers, heat transport is dominated by high thermal diffusivity, therefor the temperature profile is assumed to move with the swimmer. Swimmer-swimmer thermal interactions are kept to minimum as the thermal conductivity of the surrounding water and adjacent cover slip are 4-10 greater than that of the oil (see Table ??). This choice of materials sets the temperature gradient (Eq. 3) to be the greatest across the tightly coupled oil droplet.

2. **The idea of effective attraction between passive particles in an active bath is not new.** There have been several investigation on activity-induced depletion interactions: D. Ray et al, *Phys. Rev. E* **90**, 013019 (2014); J. Harder et al, *J. Chem. Phys.* **141**, 194901 (2014); F. Smallenburg et al, *Phys. Rev. E* **92**, 032304 (2015). These papers should be a least mentioned. They also show that depletion interactions can be repulsive which would destroy the picture proposed by the authors. Can the authors measure the effective force between a pair of peanut particles to see the basic effect on the particle pair scale? While I agree that the depletion picture is a good one, it should be supported with more data. We thank the reviewer for drawing our attention to relevant past numerical simulations done in the field. We added the following mentioning in the text to refer the readers: Previous numerical simulations suggested that at low concentrations, active particles may induce an effective interaction between a pair of passive particles through what is known as Active Casimir Effect^{???}. There are a couple of qualitative differences in the setup of the simulations and our experimental findings which are worth mentioning:

- (a) The mentioned simulations consider the case of only two passive particles at a fixed separation submerged in a bath of active particles much smaller than the passive particles. This setting is very different than the one we are treating where the active and passive particles are of the same order both in size and in concentration. Moreover, it is interesting to note that we expect a minimal concentration of the passive particles to trigger the corralling. In section 7 in the SI we mention sub-linear contribution of the perimeter of a crystal of N_P passive particles to the excluded volume (scaling as $\sqrt{N_P}$). This suggests that a minimal number of passive particles is required for the overlap of the excluded areas to be significant. Indeed we did not see corralling with very small number of passive particles. It would be interesting to further investigate the finite size effect both experimentally and in simulations.
- (b) The corralling we observe happens at relatively high area fractions ($\phi \approx 0.4$) similar to the area fraction where MIPS is found (and theoretically predicted, see Refs. 34-38). By contrast the simulation papers mentioned, explicitly treat the low concentration regime of the active bath (see “dilute suspension” in Smallenburg *et al*, “volume fraction of 0.1” in Harder *et al*, and “we consider the dilute limit” in Ray *et al*). At this limit, the MIPS is not expected to play a significant role in the dynamics. At such low concentrations we indeed observed active particles forming and breaking passive clusters, but these did not conclude in the large scale corralling seen at high

concentrations.

It is also worth noting a distinct difference in the points of view of the mentioned references and ours. Previous work tried to find a thermodynamic description of an active system, whereas we treat the micro-swimmers as a dynamical system. In the mentioned theoretical work, an effective potential (attractive or repulsive) between the passive particles is placed at the focal point. A construction of such an effective potential would have been very beneficial, as it could be used in the construction of a thermodynamic model. An instructive example from equilibrium thermodynamics for this is depletion interaction, where the presence of many small particles (depletants) can be mapped into an effective pairwise attractive potential between large particles (once accounting for their excluded volume). This effective pairwise attraction can be expressed as a free-energy. In our work we do borrow the idea of excluded volume found in depletion, we do not however pursue a construction of a pairwise potential, as it is not clear if such a potential can be derived for a high-concentration, out-of-equilibrium system (see Ref 39). Instead, we treat the system as a dynamical system characterized by an instability (the critical MIPS concentration). The passive particles take the role of triggering this instability simply by occupying space and reducing the available area.

- 3. In the methods part it is stated that dimers were synthesized by binding a light absorbing particle to a fluid oil particle. The mechanisms of binding should be explained in more detail.**

We agree with the reviewer that the previous version of the manuscript lacks a proper description of the binding mechanism. To address this we added 4 new subsections in the Methods section (“Synthesis Overview”, “Emulsion Synthesis”, “Coupling Light Absorbing Particles to Liquid Droplets”, and “Dimer Purification”). The new sections are also supplemented with Figures 5, 6, 7, and 8, including schematics as well as photos of actual experimental steps. We note that the manuscript does refer the reader to previous work focusing on the synthesis (see Ref 24 in the manuscript) where the chemical process and its variations along with kinetic analysis, and throughput-yield characterization are discussed in great detail. We appreciate the reviewer’s mentioning of the issue, and we believe that in its current version, both the mechanism and the protocol can be easily adapted by an inspired researcher interested in extending the work on this system.

- 4. The insets of Figure 3B and 3D are invisible. I wonder why the displacements in 3B for $\phi < \phi^*$ are smaller than that for $\phi > \phi^*$ but maybe this is due to the poor resolution of**

the figure.

We thank the reviewer for pointing out the poor visibility of the inset in Figs. 3b and 3d, and we improved the Figures in the current version. We refer the reviewer to item 6 in the response to all reviewers for a detailed description of the updated version. We thank the reviewer for pointing out the resulting ambiguity and believe the current version is clearer.

5. In the movies trimers and spinners are seen, is there any comment on those?

The reviewer correctly points out that some of the particles observed are different than the typical dimer structure. Indeed some of the swimmers have two droplets or two light absorbing particles changing the nature of their motility (for example, making them spin). The concentration of these peculiar swimmers is however very low ($\leq 3\%$) and do not seem to play a significant role in the collective effects observed. It would be interesting to further research their properties. For example, we found that trimers having two light absorbing particles on a single droplet swim much faster than a dimer (consistent with our model).

6. typo on page 3: Suuplementary

Fixed. Thanks.

Reviewer # 2

The authors introduce a novel type of synthetic polar active particle by coupling a light absorbing colloid to a fluid droplet, resulting in a surface-tension driven propulsion mechanism. In contrast to chemically powered active colloids, this new experimental microswimmer system enables the realization and study of long-lived coherent non-equilibrium dynamics at high density.

The authors explain and characterize the dynamics of individual swimmer in terms of a thermocapillary model, and they observe interesting collective dynamics at intermediate and high filling friction. The authors further analyze mixtures of active and passive colloids, and find that their experimental observations agree with theoretical predictions based on simple geometric considerations (Eq.2, Fig.4)

The paper is well written, the experimental characterization is comprehensive, and the theoretical arguments appear convincing.

More generally, I believe that this new synthetic active particle system provides an interesting alternative to chemically powered colloidal systems. I am therefore happy to recommend this work for publication, but would ask the authors to consider and address the minor questions below.

We are delighted that the reviewer is happy to recommend our work for publication.

Response to comments

1. **Fig 1B,C: What is the duration of the trajectories shown in these two panels, respectively?**

In both cases their durations are **8 seconds**. We added the following remark in the caption of the figure to clarify this point: “8 second long trajectories of individual swimmers show transition from diffusive dynamics when light is off (b) to ballistic when light is turned on (c) (see Supplementary Video 3)”.

Thanks.

2. **Fig.3: The segregation into a long lived state of coexisting dilute and dense fluid phase seems reminiscent of the phenomenology predicted in Fig. 2(d) of Phys Rev E 89: 010302(R), 2014.**

We thank the reviewer for pointing us to the work by Wensink *et al* which we added in the revised version of the manuscript. In their work they observed many of the effects we see (cluster formation,

motion of active particles *inside* dense region, as well as segregation of mixed populations). This work is particularly relevant to our experimental system. We have focused on peanut shape particles, but our synthesis allows for controlling the relative size of the liquid droplet (front) to the light absorbing particle (rear). This may prove as a synthetic route for forming swimmers with controlled fore-aft asymmetry as proposed by the reviewer (see Ref 24). We added the following in the main text to highlight this point: *We demonstrate that the synthetic route makes swimmers of different materials accessible, with tuneable fore-aft geometrical asymmetry, making previous numerical work experimentally realizable* .

3. I am not entirely convinced by the “cooperation angle” of the presentation. The interesting collective dynamics reported in the paper arises mostly from basic physical interactions that do not seem to meet the criteria of “cooperation” in more complex biological systems.

Here we chose to adhere to a definition for cooperation borrowed from the field of swarm-robotics (see Ref 20): a swarm is said to cooperate when an increase in its size displays a “*superlinear performance increase*” in its ability to execute a task. This is exactly the case for our swimmers — at low concentrations the swarm does not corral passive particles at all. When the swarm’s size is increased past the critical concentration (as predicted in Eq. 2) corralling is observed. This transition from no-corralling to corralling is superlinear. The reviewer correctly identifies that ”cooperation” may be treated differently across different scientific disciplines (be it biology[?], anthropology[?], or economy[?]). In our work we propose that cooperation can effectively emerge even from very simple ingredients — a critical concentration and excluded area. Could it be that a similar minimal model can predict cooperation in higher forms of life? We hope our findings will further inspire this very important, on-going, interdisciplinary discussion.

Minor comments:

In brackets below Eq (1): remove “droplet”? Ref [2] has wrong title and page number Ref [5] has incorrect page/article number Ref [13] has incorrect page/article number Ref [18] has incorrect page/article number Refs [33-36,39,41] have incorrect page/article number

Response to minor comments:

1. Index f refers to the the fluid droplet.
2. Corrected title in Ref [2].
3. Corrected page/article number in Refs [2,5,13,18,33-36,39,41]. Thanks.

Reviewer # 3

The authors present a new type of microswimmers that is based on a fuel-less non-phoretic propulsion mechanism, where the particles are supplied with energy by light. The main advantage of the new propulsion mechanism is that the particles remain motile also in highly crowded assemblies. Furthermore, the authors realized the particles in experiments and found that they show an interesting cooperative motion with parallels to swarming of animals.

The findings are certainly interesting for the large active matter community. A type of active particles that behaves more like bacteria or animals than the various phoretically propelled active particles is indeed useful. Artificial active particles are quite often compared to motile microorganisms and animals and these new particles could probably be applied as models for bacteria and animals.

The paper is well written and the presentation is clear.

However, I cannot recommend publication of the paper in its present form. The authors should address the issues listed in the following. Some problems can easily be solved by the authors. Other issues require to provide more information in the manuscript on which basis it will become clear whether the findings are sufficiently relevant for publication in the chosen journal.

We are happy that the reviewer found the paper well written and clear, and that the findings are interesting for the large active matter community. We hope we were able to address all the referees concerns in our response below.

Response to comments

1. **The propulsion mechanism of the proposed particles should be explained in more detail.**

It would be very helpful if the authors provide a sketch illustrating clearly how it works.

We agree with the reviewer and in order to better illuminate the thermo-capillary mechanism, we added a diagram in Fig.2b, and the following explanation in the caption of that figure: “Inset illustrates the fluid-flow where the surface-tension gradient drives the liquid-liquid interface to re-circulate (black) and propel the swimmer through the hydrodynamic coupling to the external flow (cyan)”. Also note that Refs 25-31 in the manuscript have previously described the mechanism of thermo-capillary motion. Especially Ref 25 (from which we adapted Eq. 1) fully solves the hydrodynamic problem for the case of a fluid sphere moving in a temperature gradient. Moreover, Sections 1.1 and 1.2 in the Supporting

Information treat the mechanism, and its measurement in great detail. We do agree with the reviewer that a graphical description in the main text was due, and believe that now the mechanism is made clear.

2. It seems that a temperature gradient caused by illumination and a resulting flow field are crucial for the propulsion of the particles. How do they differ from the existing thermophoretically propelled particles studied, for example, by Cichos (Leipzig)?

The reviewer correctly points out that thermo-phoresis and thermo-capillary are intimately linked. Indeed, early in the development of the new swimmer-system we asked ourselves a similar question — how can we empirically discriminate between the two mechanisms? An insightful work by Ruckenstein addressed this very link in a paper entitled “Can Phoretic Motions Be Treated as Interfacial Tension Gradient Driven Phenomena?”[?]. There, it is found that in both cases the speed of the particle depends on the viscosity of the outer liquid, but *the speed of the thermo-capillary swimmer has a strong dependence on the viscosity of the inner fluid* (η_f in Eq. 1). This dependence is of course absent for a *solid*, thermo-phoretic, particle. To resolve this ambiguity, we measured the swimming speed for swimmers with droplets of different internal viscosity. Using swimmers made of a homologous series of alkanes allowed to tune the droplets’ internal viscosity while keeping important material properties nearly constant (refractive index, thermal-conductivity, interfacial surface-tension and so on). We synthesized and measure speeds of swimmers with droplets’ viscosities ranging over an order of magnitude. Our findings are summarized in Fig.2f in the manuscript, showing a clear decrease of the speed with increasing inner viscosity. This agreement became more evident when we computed the theoretically expected speed (solid curve on same plot). This curve has *no fitting parameters* and only uses Eq.1, of which parameters were found empirically, or taken from the literature (see Sections 1.1-1.4 in SI as well as Eq. 4-5 therein, and Table 1). We thank the reviewer for stressing this point and we updated the manuscript, along with a citation of Ruckenstein’s work: *By synthesising a family of swimmers made from a homologous series of alkanes, we observe a decrease in swimming speed, v , with increasing internal viscosity, η_f , as required by a thermo-capillary mechanism (Fig.2f)[?]. When accounting for the reduced mobility of particles near a solid surface[?], we find quantitative agreement between measured swimming speed, and the light driven thermo-capillary model for swimmers of different compositions (see Fig.2f) with no fitting parameters.*

3. Second paragraph, line 4: Replace ”homosapiens” by ”homo sapiens”.

Done. Thank you.

4. **The authors mention collisions of the particles. The particles, however, consist of a combination of a solid particle and a droplet. Why do they stay intact after a collision? Why do the droplets not coalesce?**

This is an excellent question. As noted on the first paragraph of the Methods section, a surfactant (SPAN 80) is used in the droplet synthesis. This is also what allows the emulsion to be stable for many months (see Methods section), despite forming a very dense cream. It should be pointed that unlike most emulsions, here the surfactant is placed *inside* the swimmer’s liquid droplet, in the so called discrete phase (typically the surfactant is in the continuous phase, *outside* the droplets). This subtle point is crucial for the drive mechanism. Swimming droplets are not new (see for example Ref 16), however previous designs had the surfactant in the continuous phase, leading to strong, long range, long duration chemical swimmer-swimmer interactions resulting in swimming slowdown. To clarify this point we added a whole section in the Methods: “Emulsion Synthesis” as well as the following remark “The suspension is then stored in a glass vial and is stable as a dense cream at room temperature for months”.

5. **The authors should comment in more detail on how the motion of the particles compares to the cooperation of microorganisms and animals. To which extent are they similar?**

We agree that it is a good idea to mention in the manuscript the greater scope of cooperation in the animal kingdom, and we added the following in the conclusions section: “The swimmers remain active at high area fractions, \$\phi_A > 0.5\$, and can be used as an experimentally tuneable model system to study a host of phenomena in dense active matter including 2D turbulence in a compressible fluid, previously seen in bacterial colonies^{???} and in groups of ants^{??}. We found that at a low area fraction (\$< 0.15\$ ), swimmers tend to quickly disperse; at intermediate area fractions (0.14 - 0.28) swimmers form transient co-moving clusters, reminiscent of the extensively studied flocking starlings, and schooling golden shiners^{??}.”

6. **Is equation (1) really applicable? It describes the velocity of a fluid droplet in a temperature gradient, but is the temperature gradient caused by heating of the small solid component of a particle really significantly larger than the particle? If yes, why is thermophoresis not relevant, which would make the particles just to another type of phoretic particles?**

The reviewer makes a good comment that can be split in two parts:

- (a) How does the spatial extent of the temperature gradient compares with the size of the heating particle?
- (b) Should the swimmers be described as thermo-phoretic swimmers?

To which we respond separately:

- (a) Eq. 1 (taken from Ref 25) describes the velocity (\vec{v}) of a fluid droplet in an external fluid when a constant temperature gradient is imposed ($\vec{\nabla}T$), giving its dependence on the diameter of the droplet (D), and the bulk properties of the two fluids (viscosity η , thermal conductivity κ , and thermo-capillary coefficient β). To leading order, the spatial extent of the magnitude of the temperature gradient is $\nabla T \propto 1/R^2$, as found by solving the heat equation of a hot sphere (now given explicitly in Eq. 3 in the revised version of the manuscript). Owing to the high thermal conductivity of water ($0.6W/mK$) and the glass cover slip ($1.4W/mK$) which are 4-10 times greater than that of the oil droplet ($0.14W/mK$) we expect the greatest temperature gradient to be found across the liquid droplet.
- (b) The main reason why we believe the mechanism of our swimmers' propulsion should be understood as thermo-capillary is the strong dependence on the viscosity of the inner fluid. Eq. 1 predicts that for a given temperature gradient, the swimming speed decreases as $1/\eta_f$ (with η_f being the viscosity of the internal fluid). Thermo-phoresis can also have a contribution to the swimming of a liquid droplet but its dependence on the viscosity of the inner droplet is much weaker (see Eq. 29 in Ruckenstein 1981[?]). In our experiments we found a strong dependence on the internal viscosity: for a given light flux, swimmers with increasing liquid droplet viscosity display decreasing swimming speed (see Figure 2f). Given the material properties of our swimmers, we find our measurement to fit quantitatively to a thermo-capillary drive.

In order to clarify the above points for the reader, we highlighted the equation for the temperature gradient (Eq. 3), and added the following text in the supporting information: “Eq. 3 shows that the temperature gradient scales as $1/R^2$ ”. We also emphasized the significance of the internal viscosity as mentioned in item 2.

7. **The figures should be improved, the quality is only average. The largest problem is the size of the labels. Sometimes they are too small and sometimes they are too large. For example, in figure 2A the numbers at the frame are much too small and in figure 3 the labels "A", ..., "D" are much too large.**

Thank you for this comment. The figure labelling has been corrected and now is consistent throughout the manuscript, making the text more readable.

8. Figure 3B, inset: Why are there two orange curves that look rather different and not just one curve?

Other reviewers raised similar concerns, we have addressed the issue and improved the inset (see items 1-5 in the general comments for details of the correction).

9. Figure 3B: Here are a histogram and a curve plotted, but there is only one vertical axis. This is strange. In particular, the histogram seems to be a probability distribution and then it makes no sense that the vertical axis has the unit μm .

Following the reviewer's comments we now see that the figure in its previous form was indeed somewhat confusing. We updated the axis and their link to the data (see item 1 in general comments). We note that it is instructive to draw the two measurements (speed-concentration curve as well as concentration histogram) on the same axes as they are intimately linked — one shows the response of the speed of an *individual* particle to its local density (displacement curve), while the other shows the resulting *collective* behaviour, leading to phase separation (histogram). This link lies at the heart of the instability known as Motility Induced Phase Separation.

10. Figure 4A: The colors red, yellow and green are explained in the figure, but the legend should explain also what the blue region means. (I know that it is explained in the caption but this is not sufficient.)

Figure 4a shows a ternary diagram and the legend refers to the *experimental outcome*, where the swarm disperses, crowds, or corrals. The different shades of the the diagram represent the *theoretical* prediction for that region. The blue region is, in a way, forbidden as it represents initial concentrations higher than hexagonal packing of spheres ($\phi > 0.91$). Since there are no experimental points in the forbidden blue region, they do not appear in the legend.

11. Figure 4, caption, line 5, middle: Where does this equation come from?

We thank the reviewer for turning our attention to the missing reference. The equation in the caption of Fig 4. comes from Eq. 2 in the main text. We now reference the equation in the caption.

12. Page 5, left column, paragraph 2, lines 2-5: I do not really understand the argument why the passive particles facilitate crowding of the active particles. It is true that they reduce the space that is available for the active particles, but on the other hand the active

particles collide with the passive ones and slow down by these collisions so that the typical crowding mechanism of active particles (motility-induced phase separation) should no longer work.

The reviewer makes a good point which we added to the manuscript: the mean motility of our microswimmers decreases at higher concentrations — this is the origin of the motility induced phase separation observed at the absence of passive particles. The reviewer correctly points out that the same happens at the introduction of passive particles — swimmers will maintain their nominal speed between collisions, but will slow down upon a collision with a passive particle. This is exactly the microscopic picture we had in mind in our model: the overall increased area fraction decreases the mean motility of the active particles. This increase can be achieved with adding active particles (as in MIPS) or by introducing passive particles (as done here). Further interesting questions may address the different extent of motility reduction by an extra active or passive particle. To lowest order, we treated the passive particles as reducing the area available for the active particles, which trigger MIPS at a concentration of active particles lower than the concentration of a pure active suspension. We then realized that at high concentrations, the spatial correlations of the passive particles must be taken into account — randomly distributed passive particles effectively occupy more area than when densely packed (as their excluded areas overlap). This striking difference is made visually clear in Fig. 13 in the Supporting Information. We added the following in the main text, making the picture more comprehensive: **An increased concentration of active particles reduces their motility, and past a critical concentration (ϕ^*) triggers MIPS. Similarly, an increased concentration of passive particles also reduces the motility of the active particles, which in turn also leads to a clustering instability. The effective increased concentration however depends on the spatial arrangement of the passive particles. Dynamically, the corraling process proceeds through two stages: 1. at early times, when the dense, passive-active, suspension is still well mixed (Fig. 4b), the ability of the swimmers to re-arrange when crowded allows them to escape the dense region (Fig. 4c-d. also see Supporting Video S6); 2. at later times, individual entrainment events are observed (see Fig. 4F and Supporting Video S2).**

- 13. The authors write that the particles form a dense fluid in the interior of the clusters and that they do not form a solid as it has been observed for phoretically propelled active particles. Does this also apply to very large clusters? In Brownian dynamics simulations of active Brownian particles that can be found in the literature, it has been observed that the particles form a solid and even crystals in the interior of a cluster. This crystallization**

occurs although the propulsion mechanism of these particles is by definition not affected by the presence of other nearby particles. Therefore I would expect that also the particles studied in the present paper show solidification in the clusters if they are sufficiently large.

The reviewer raises a good question: Would very large clusters also have a fluid interior? Based on our experimental observations, we believe that the answer to this is positive. Supporting Video S1 shows a cluster of about **100** swimmers with internal, fluid-like motion (highlighting individual particles coming in and out of the dense region). Supporting Video S5 shows a larger dense cluster of about **400** particles, still showing internal flows. To further facilitate this point, we attach here another experiment (not found in the manuscript) with a cluster of about **1000** particles that *still shows strong internal dynamics*. We suspect that the liquid nature of leading front of each swimmer, allows it to move even in a crowded environment. Friction has been shown lately to be important even at the colloidal scale (Phys. Rev. Lett. 112, 098302), giving rise to shear thickening. Having a liquid front helps fluidize the dense cluster, thus avoiding solidification. To further clarify this point in the manuscript, we added the following in the main text: Even crowded regions of over 1000 swimmers display a turbulent flow.

14. The journal name of reference [15] is wrong.

We fixed the journal name. Thank you for noting that.

15. Reference [22] is incomplete.

We fixed the reference and now it is more complete. Thank you.

16. Acknowledgments, line 6, middle: ", and), and" is wrong.

We fixed the typo in the acknowledgement. Thanks.

17. Table 1 does not look good. At least a horizontal line between rows 1 and 2 is needed.

We added a horizontal line to Table 1, and find that indeed the table is clearer this way. Thank you.

Reviewer # 4

This paper reports the development of a new class of colloidal microswimmers that promise access to dense regimes of active fluids that have heretofore not been able to be accessed in synthetic systems. The propulsion in the high density regime is accomplished by using light to propel the particles rather than a fuel that would be diffusion limited in crowded environments. Further this allows nice control of local velocity as a function of temperature gradient. Several well known results from the self-propelled particle literature are nicely validated here including MIPS. The active-passive phase separation system is underexplored but serves as a further validation of the potential for this system to explore the phenomenology of self-propelled particles in a well controlled system. I like it and have no criticism for the authors. All the active matter people will be looking forward with bated breath to further work on this system. We are humbled by the reviewer's praises and share the hope that the system and findings presented herein, will help propel forward the field of active matter.

References

- [1] Ray, D., Reichhardt, C. & Reichhardt, C. J. O. Casimir effect in active matter systems. *Physical Review E* **90**, 013019 (2014). URL <https://link.aps.org/doi/10.1103/PhysRevE.90.013019>. 1402.6372.
- [2] Harder, J., Mallory, S. A., Tung, C., Valeriani, C. & Cacciuto, A. The role of particle shape in active depletion. *Journal of Chemical Physics* **141** (2014). 1407.6743.
- [3] Smallegang, F. & Löwen, H. Swim pressure on walls with curves and corners. *Physical Review E* **92**, 032304 (2015). URL <https://link.aps.org/doi/10.1103/PhysRevE.92.032304>. 1504.05080.
- [4] Wensink, H. H., Kantsler, V., Goldstein, R. E. & Dunkel, J. Controlling active self-assembly through broken particle-shape symmetry. *Physical Review E* **89**, 010302 (2014). URL <https://link.aps.org/doi/10.1103/PhysRevE.89.010302>.
- [5] Attanasi, A. *et al.* Information transfer and behavioural inertia in starling flocks. *Nature Physics* **10**, 691–696 (2014). URL <http://www.nature.com/articles/nphys3035>.
- [6] Harari, Y. N. *Sapiens : a brief history of humankind* (Harper, New York, 2015).
- [7] Marx, K. Capital: A Critique of Political Economy. In *Part 4*, chap. 13 (1887).
- [8] Ruckenstein, E. Can phoretic motions be treated as interfacial tension gradient driven phenomena? *Journal of Colloid and Interface Science* **83**, 77–81 (1981). URL <https://linkinghub.elsevier.com/retrieve/pii/0021979781900114>.
- [9] Happel, J. & Brenner, H. *Low Reynolds Number Hydrodynamics* (Springer, Englewood Cliffs, 1983).
- [10] Xu, H., Dauparas, J., Das, D., Lauga, E. & Wu, Y. Self-organization of swimmers drives long-range fluid transport in bacterial colonies. *Nature Communications* **10**, 1–12 (2019). URL <http://dx.doi.org/10.1038/s41467-019-09818-2>.
- [11] Dunkel, J. *et al.* Fluid Dynamics of Bacterial Turbulence. *Physical Review Letters* **110**, 228102 (2013). URL <https://link.aps.org/doi/10.1103/PhysRevLett.110.228102>. abs/1302.5277.
- [12] Wensink, H. H. *et al.* Meso-scale turbulence in living fluids. *Proceedings of the National Academy of Sciences* **109**, 14308–14313 (2012). abs/1302.5277.

- [13] Zhang, H. P., Be'er, A., Florin, E. L. & Swinney, H. L. Collective motion and density fluctuations in bacterial colonies. *Proceedings of the National Academy of Sciences of the United States of America* **107**, 13626–13630 (2010).
- [14] Dombrowski, C., Cisneros, L., Chatkaew, S., Goldstein, R. E. & Kessler, J. O. Self-Concentration and Large-Scale Coherence in Bacterial Dynamics. *Physical Review Letters* **93**, 098103 (2004).
- [15] Altshuler, E. *et al.* Symmetry Breaking in Escaping Ants. *The American Naturalist* **166**, 643–649 (2005). URL <http://www.journals.uchicago.edu/doi/10.1086/498139>.
- [16] Gelblum, A. *et al.* Ant groups optimally amplify the effect of transiently informed individuals. *Nature Communications* **6**, 7729 (2015). URL <http://www.nature.com/articles/ncomms8729>.
- [17] Katz, Y., Tunstrom, K., Ioannou, C. C., Huepe, C. & Couzin, I. D. Inferring the structure and dynamics of interactions in schooling fish. *Proceedings of the National Academy of Sciences* **108**, 18720–18725 (2011). URL <http://www.pnas.org/cgi/doi/10.1073/pnas.1107583108>.

REVIEWERS' COMMENTS

Reviewer #1 (Remarks to the Author):

The authors have revised their manuscript carefully taking my remarks into account. I now recommend publication in Nature Communications.

Reviewer #2 (Remarks to the Author):

In the revised manuscript version, the authors have satisfactorily addressed my comments and questions as well as those of the other reviewers. This has further strengthened the overall presentation and the main conclusions. As already stated in my previous report, this is a very interesting study of a novel experimental active matter systems.

I am therefore happy to strongly recommend this revised version for publication in Nature Communications.

Reviewer #3 (Remarks to the Author):

The authors have answered all my questions in detail and significantly improved the manuscript.

I am happy with all changes except for the style of the figures. Their quality is still below what one can expect for good journals.

Some examples:

- Fig. 1-4, 7 and 8: The size of the labels (symbols or numbers) in the figures is usually very different from the normal font size (= font size of the main text). Furthermore, the size often varies a lot within a single figure. However, one should use normal font size for all labels in all figures.

- Fig. 4: I suggest to write something like "forbidden" into the blue area in plot (a) to make clear what the blue area means. It should be clear also without reading the caption.

This change is optional.

- Fig. 7: The small labels are much too small. On a printout of the paper, I cannot read these labels.

I think that these problems with the labels of the figures will be solved by the journal before publication.

Therefore, I now recommend to accept this work for publication in Nature Communications.